# Exploring the Health Literacy and Patient Activation Among Patients with Glaucoma: A Cross-Sectional Study

**DOI:** 10.3390/clinpract15020024

**Published:** 2025-01-23

**Authors:** Lοukia Tsichla, Evridiki Patelarou, Efstathios Detorakis, Miltiadis Tsilibaris, Athina Patelarou, Antonios Christodoulakis, Eleni Dokoutsidou, Konstantinos Giakoumidakis

**Affiliations:** 1Department of Nursing, School of Health Sciences, Hellenic Mediterranean University, 71410 Heraklion, Greece; ddk192@edu.hmu.gr (L.T.); epatelarou@hmu.gr (E.P.); apatelarou@hmu.gr (A.P.); christodoulakisa@uoc.gr (A.C.); 2School of Medicine, University of Crete Medical School, Voutes, 71110 Heraklion, Greece; edetorak@uoc.gr (E.D.); tsilimb@med.uoc.gr (M.T.); 3Department of Nursing, University of West Attica, 12243 Athens, Greece; edokout@uniwa.gr

**Keywords:** health literacy, glaucoma, self-management, patient activation, cross-sectional

## Abstract

**Background:** Glaucoma is one of the leading causes of blindness that can be mitigated through early recognition and effective management. Specifically, early and effective self-management outside hospitals can slow disease progression and reduce its negative daily impact. This includes adherence to medication, high levels of health literacy (requires patients to be able to find, understand, and use relevant health information), early recognition of symptoms, regular visits to ophthalmologists, etc. However, there is a lack of empirical evidence regarding levels of adherence to medication and health literacy in glaucoma patients in Greece. This study aimed to assess health literacy and self-management activation levels in glaucoma patients and explore the relationship between these factors. **Materials and Methods:** A total of 312 glaucoma patients were recruited from outpatient ophthalmology clinics in Heraklion, Greece, between November 2023 and May 2024 through convenience sampling. The Greek versions of the Patient Activation Measure-13 (PAM-13) and the European Health Literacy Survey Questionnaire 16 (HLS-EU-16) evaluated self-management activation and health literacy, respectively. **Results:** Patients exhibited low self-management activation (level 1: disengaged and overwhelmed, =40.7, SD: ±29.9) and sufficient health literacy (=10.7, SD: ±3.7). No significant association was found between health literacy and self-management activation (*p* = 0.602). **Conclusions:** Glaucoma patients had low to moderate levels of self-management activation and health literacy, without a significant association between them. Therefore, multifaceted strategies are needed to enhance these patients’ activation and health literacy. Further research, using larger, multi-center samples, is needed to clarify the link between these variables.

## 1. Introduction

Glaucoma, one of the most important chronic diseases, is a major public health problem due to its serious impact on the vision and quality of life of patients [1]. It is characterized by progressive loss of vision and increased intraocular pressure. These factors make early diagnosis and proper management essential. According to the American Academy of Ophthalmology (2021), early and effective self-management of the disease outside hospitals can slow its progression [2]. Additionally, it can reduce its negative impact on patients’ daily lives. This includes adherence to medication, early recognition of symptoms, regular visits to the ophthalmologist, and proper use of eye drops [2]. However, many patients find it difficult to adequately manage their condition. This is due to several factors, including health literacy and patient activation [3].

Health literacy plays a key role in managing chronic diseases [4]. Moreover, comprehensive daily disease management requires patients to find, understand, and use relevant health information [5]. The literature highlights that health literacy depends not only on individual skills but also on health system accessibility, healthcare providers’ communication skills, and the complexity of health information. Additionally, it is influenced by healthcare providers’ communication skills and the complexity of health information [4]. Patients with a high level of health literacy are better able to understand healthcare professionals’ instructions. They can also read and comprehend drug labels, follow treatment plans correctly, and make informed health decisions. In contrast, patients with a low level of health literacy face significant difficulties. These include understanding healthcare professionals’ instructions, making errors in taking medications, and experiencing worse health outcomes, such as increased hospitalization rates and higher mortality [6].

The level of patient activation is an equally important factor in the management of chronic conditions. According to Hibbard et al. (2005), patient activation refers to the extent to which patients possess the knowledge, skills, and confidence to manage their health [1]. More specifically, patients with higher activation levels are more likely to adhere to treatment instructions, adopt healthy behaviors, and avoid unnecessary hospitalization [7]. On the other hand, patients with lower activation levels tend to be more passive. They often face greater challenges in the self-management of their health. Therefore, interventions aimed at enhancing patient activation involve educating patients, providing clear and accessible information, empowering them with support, and equipping them with self-management tools [8].

Although health literacy and patient activation have been found to be uncorrelated, various factors may mediate the relationship between these variables in chronic disease management. These factors may influence either the understanding of health-related information or the willingness of patients to actively participate in their care. Specifically, socioeconomic factors such as educational attainment and income significantly impact the understanding and utilization of health information [9,10]. Moreover, self-efficacy and psychological conditions, including anxiety or depression, are closely associated with patients’ readiness to assume an active role in their healthcare [11]. Additionally, communication with healthcare professionals and support from social networks can enhance patient engagement by providing a supportive environment for decision-making [12]. Lastly, access to health technologies and digital applications can empower patients, whereas information overload may act as a barrier [13]. Understanding these mediating factors is critical for developing tailored interventions in chronic disease management.

In Greece, the levels of health literacy and patient activation remain low [14], which makes it difficult to manage chronic diseases such as glaucoma [15]. According to a recent study, the lack of appropriate education and support programs for patients with chronic diseases has led to increased rates of non-adherence to treatment guidelines and worse health outcomes [16].

This study aimed to assess the health literacy and self-management activation levels of glaucoma patients in outpatient clinics and to explore the relationship between these factors. Despite extensive research, there is still a significant gap in the literature regarding the interaction between health literacy and patients’ levels of activation in glaucoma management [17]. This study intends to contribute new insights to the existing body of knowledge.

## 2. Materials and Methods

### 2.1. Study Design and Variables

A cross-sectional study was conducted in 312 patients with glaucoma from an outpatient clinic of the University Hospital in Heraklion, Crete, Greece. In the present study, the dependent variables were the level of health literacy and the level of activity of glaucoma patients, while the independent variables included biological sex, age, educational level, and comorbidities (i.e., the presence of any additional chronic health conditions that may affect the management and progression of glaucoma), including diabetes mellitus, hypertension, cardiovascular diseases, cancer, and psychological disorders.

The population of the present study consisted of patients diagnosed with glaucoma. The diagnosis of glaucoma was based on previously published diagnostic criteria: (a) high intraocular pressure, (b) optic-nerve atrophy, and (c) deficits in the visual fields [18]. The diagnosis of glaucoma was confirmed using these criteria, indicating that the disease was not in an early stage, as evidenced by extensive visual field defects (MD > −8.5 dB) [18].

### 2.2. Participants and Sample

Τhe sample was obtained through convenience sampling of glaucoma patients who attended the outpatient clinic of the Ophthalmology Department of the University Hospital of Heraklion (Crete, Greece) between November 2023 and May 2024. The inclusion criteria were as follows: (i) age ≥ 18 years, (ii) informed written consent to participate in the study, (iii) adequate knowledge of the Greek language (writing and reading), and (iv) a confirmed diagnosis of the condition. Patients with incomplete information in their health records or those who provided incomplete information were excluded.

### 2.3. Data Collection and Instruments

Data were collected using three questionnaires: a self-report questionnaire for demographic characteristics (biological sex, age, educational level, and comorbidities) specifically designed for this study, the Greek version of the European Health Literacy Survey Questionnaire 16, and the Greek version of the Patient Activation Measure 13. It should be noted that written permission was obtained to use each questionnaire from its respective developer.

#### 2.3.1. European Health Literacy Survey Questionnaire 16 (HLS-EU-16)

The European Health Literacy Survey Questionnaire 16 (HLS-EU-16) [19] was used to assess participants’ health literacy. This tool has been translated into Greek and validated within the Greek population [20]. The questionnaire is available in three versions, depending on the number of questions (47, 16, or 6). The version used for this study was developed based on the Rasch model [19].

It includes 16 questions, each of which is answered on a 4-point Likert scale ranging from 1 (“very easy”) to 4 (“very difficult”). Answers categorized as “very easy” and “easy” receive one point, while “very difficult” and “difficult” answers do not receive a point [19]. To calculate the total score of the questionnaire, the score for each answer is summed, with the final score ranging from 0 to 16 [19]. Participants are then categorized based on their scores: specifically, scores between 0 and 8 indicate inadequate health literacy, scores of 9–12 suggest problematic health literacy, and scores of 13–16 indicate sufficient health literacy [19].

#### 2.3.2. The Patient Activation Measure-13 (PAM-13)

The Patient Activation Measure-13 (PAM-13) questionnaire was used to assess the patient’s activity level [21]. Additionally, this tool has been translated into the Greek language and validated within the Greek population [21]. It was created by Hibbard JH [22], as a short version of the original questionnaire with 22 questions [1]. The PAM-13 includes 13 questions, each of which is answered on a 4-point Likert scale ranging from 1 (“strongly disagree”) to 4 (“strongly agree”), with the additional option “not applicable”. To calculate the total score of the questionnaire, the score of each answer is summed and the sum is divided by the number of answers given by the study participant, excluding the “not applicable” answers [21]. This calculation results in a “raw” score ranging from 13 to 52. This “raw” score is then algebraically transformed to a standardized score between 0 and 100. Higher PAM score values are indicative of higher patient activity levels [23,24]. Furthermore, patients based on their scores are classified into four levels of activity: (a) level 1 with a score of ≤47.0 (“disengaged and overwhelmed”), (b) level 2 with a score of 47.1–55.1 (becoming aware but still struggling), (c) level 3 with a score of 55.2–67.0 (taking action and gaining control), and (d) level 4 with a score ≥67.1 (maintaining behavior and pushing further). Moreover, it has been used primarily in patients with chronic diseases but also at the level of primary healthcare. More specifically, it has been evaluated in a series of patients with chronic diseases, elderly patients with comorbidities, patients with surgical health problems, and patients suffering from neurological diseases, diabetes mellitus, and osteoarthritis [23].

### 2.4. Statistical Analysis

We performed the statistical analysis using IBM SPSS version 26.0. Continuous variables were expressed as mean ± standard deviation, and categorical variables were expressed as frequency and percentage. Internal consistency was assessed using Cronbach’s alpha. A Cronbach’s α coefficient >0.7 indicates acceptable reliability, suggesting that the items are interdependent and homogeneous in terms of the construct they measure. To test the condition of normality, the Shapiro–Wilk test was applied, along with graphical methods such as the “Normal Q-Q plot”, “Detrended Normal Q-Q plot”, and “Box Plot”. To correlate two continuous variables, we used the Pearson coefficient (r). Spearman’s correlation index (ρ) was employed to investigate the relationship between a continuous and an ordinal variable. For the relationship between a continuous and a dichotomous variable, the Point-Biserial coefficient (pbs) was used. Moreover, a scatter plot matrix was created to illustrate the bivariate relationships among various combinations of variables. Each scatter plot within the matrix depicts the correlation between a specific pair of variables, thereby facilitating the examination of multiple relationships within a single visual representation. For all tests, statistical differences were considered significant at *p* < 0.05.

## 3. Results

The study sample consisted of 312 patients, 134 women (42.9%) and 178 men (57.1%), with a mean age of 63.9 ± 14.4 years; 228 patients (73.1%) had primary educational levels, 44 (14.1%) received secondary, and 40 (12.8%) had tertiary educational levels. Two hundred and eighty-seven patients (92%) had concomitant disease (Table 1). Cronbach’s alphas for HLS-EU16 and PAM-13 were 0.88 and 0.95, respectively.

Regarding the activity level of the patients, the mean value of the PAM-13 questionnaire was calculated as 40.7 out of 100 (±23.9). As shown in Table 2, 211 patients (67.6%) were classified as level 1 (disengaged and overwhelmed), 29 (9.3%) as level 2 (becoming aware but still struggling), 17 (5.4%) as level 3 (taking action and gaining control), and 55 (17.6%) as level 4 (maintaining behavior and pushing further).

There was no statistically significant correlation between the HLS-EU16 and PAM-13 (r (312) = −0.030, *p* = 0.602), as depicted in Table 3.

Additionally, there was no statistically significant correlation between PAM-13 and age (r (312) = 0.028, *p* = 0.617). However, there was a statistically significant, though low, positive correlation between PAM-13 and educational level (ρ (312) = 0.122, *p* = 0.031). No significant correlation was found between PAM-13 and comorbidity (pbs (312) = −0.039, *p* = 0.492). Additionally, HLS-EU16 showed no statistically significant correlation with biological sex [pbs (312) = 0.007, *p* = 0.903] and age [r (312) = 0.060, *p* = 0.291] (Table 4).

Table 5 delineates the results of Spearman’s correlation analysis exploring the interrelations among educational level, PAM-13 scores, and HLS-EU16 scores. A statistically significant, albeit weak, positive correlation was identified between educational level and PAM-13 scores (ρ = 0.122, *p* = 0.031), indicating that higher educational attainment correlates with enhanced patient activation as quantified by the PAM-13 scale. However, no significant correlations were found between HLS and educational level (*p* > 0.05).

The participants’ average health literacy level was 10.7 out of 16 (SD: ±3.7). As highlighted in Table 6, of the participants, 79 (25.3%) had inadequate health literacy, 109 (34.9%) had problematic literacy levels, and 124 (39.7%) demonstrated sufficient literacy.

Figure 1 shows, by pairs, the relationship between the three variables: age, PAM 13, and HLS.

## 4. Discussion

The present study aimed to assess the health literacy and self-management activation levels of glaucoma patients in outpatient clinics and to explore the relationship between these two factors. Our findings suggest that most participants had low levels of patient activation and health literacy, ranging from insufficient to problematic. Notably, there was no significant correlation between health literacy and self-management activation. Moreover, no significant associations were found between patient activation and demographic factors such as sex or age, suggesting that these variables do not influence activation levels. However, we observed a statistically significant, though weak, positive correlation between patient activation (PAM-13) and educational level.

A major finding of the present study was that patients with glaucoma exhibited insufficient and problematic levels of health literacy. Additionally, their activation levels were categorized at the lowest stage, level 1, indicating that they were disengaged and overwhelmed. This suggests that these patients may not adequately understand the information provided to them regarding their condition and its management. Health literacy is a crucial public health goal as it directly affects individuals’ ability to manage their health and make informed decisions [25]. The lack of this literacy can seriously affect patients’ health, leading to poor adherence to medical instructions and, consequently, worsening their health condition [26].

Another major finding of the present study was that patients’ activation levels were at the lowest level, characterized by disengagement and feeling overwhelmed. This is problematic since patients with low activation levels are more likely to feel overwhelmed and less likely to engage actively in their healthcare [27]. This means that these patients feel powerless in the face of their illness and do not take appropriate initiatives to manage their condition. These findings suggest the need for better education and support for patients with glaucoma to improve their health literacy and activation levels. Increasing patient activation levels can reduce healthcare costs and lead to better health outcomes [28].

As previously mentioned, another key finding of this study was the absence of a significant association between the patient’s health literacy level and their activation level. Some might expect a rational relationship between these two variables, given that improved health literacy typically leads to better knowledge and understanding of the disease. This knowledge forms a crucial foundation for developing self-management and self-care behaviors, both of which inherently involve active participation. Conversely, and in line with our results, the literature describes studies that do not show a clear relationship or dependence between health literacy and patient activation levels [29]. It seems that while health literacy is crucial, it does not always translate into higher patient activation. However, studies that have evaluated the association between health literacy and patient activation are limited or contradictory regarding the link between these two factors [30]. Additionally, a review highlighted that the relationship between health literacy and patient activation is not always straightforward and may be influenced by other factors, such as socioeconomic conditions and support from the healthcare system [31]. A potential explanation for this could be that other factors, such as cultural differences or the quality of information provided, may have a more significant influence on the understanding and use of health information. Furthermore, patients’ low activation despite adequate health literacy may suggest the need for further education and support to enhance their involvement in disease management. Support programs, expert guidance, and the provision of specific strategies can significantly enhance activation [32].

The finding that health literacy did not show a statistically significant correlation with education level contrasts with older studies, which often indicate that education enhances an individual’s ability to understand and use health information [16,31]. However, more recent studies show that the complexity of health information can be a barrier even for people with higher education, as specialized medical terminology and concepts can pose challenges regardless of the general level of education [33]. Additionally, a study has suggested that cultural, linguistic, and contextual factors can significantly impact health literacy [34]. For instance, individuals with higher education but who are not proficient in the language of healthcare materials may have lower health literacy [35]. Moreover, the relationship between educational level and patient activation is consistent with the literature, which suggests that more educated patients are generally more engaged in managing their health. Studies show that patients with higher levels of education are more likely to be active and involved in their healthcare [27] and that patient activation, influenced by education level, is associated with better health outcomes [13]. Although knowledge of health information does not always translate into the application of this knowledge in daily decisions—due to a gap between understanding health information and being able or willing to act based on that knowledge [36]—the association mentioned above is reasonable. A higher education level is more likely to be associated with better patient activation, as individuals with more education may be better equipped to engage in self-care and manage their health effectively. Furthermore, while patients may have the required knowledge, they might lack the tools or support needed to translate that knowledge into action.

Patient activation levels and comorbidities were not significantly associated in our study. This could suggest that glaucoma patients, due to severe comorbidities, often have reduced compliance or interest in managing their glaucoma. This may lead to inertia in terms of monitoring and treating the condition, thereby increasing the risk of ocular health deterioration. In addition, the lack of a significant association between health literacy levels and comorbidities may indicate the need for patients to be better informed due to the complexity of their condition. Moreover, it underscores the necessity for targeted nursing educational interventions in outpatient clinics for glaucoma patients, taking into account the presence of comorbidities and the complex nature of health management in these patients. Furthermore, the lack of a statistically significant correlation between health literacy and demographic factors in this study aligns with findings from other studies, which also highlight that health literacy alone may not address all barriers to effective patient engagement [37,38].

In the present study, Spearman’s correlation analysis was conducted to examine the relationships between educational level, the PAM-13 score, and the HLS-EU16 score. A statistically significant, albeit weak, positive correlation was found between educational level and the PAM-13 score (ρ = 0.122, *p* = 0.031), suggesting that higher educational attainment is associated with increased patient activation, as quantified by the PAM-13 scale. However, no significant correlations were found between HLS and educational level (*p* > 0.05). The results of this analysis suggest that while educational level may influence patient activation, it does not show a substantial correlation with health literacy, as assessed by the HLS-EU16 scale in this specific sample. These findings suggest that while educational level may in-fluence patient activation, it does not show a substantial correlation with health literacy, as assessed by the HLS-EU16 scale in this specific sample.

These findings have significant implications for both patients and healthcare professionals. It is important for healthcare professionals to actively encourage patients to engage in their own care, seek information about their condition, and participate in decision-making. Overcoming challenges related to treatment adherence is crucial, and this requires personalized educational interventions that take into account the patient’s needs, cultural background, and level of understanding [39,40]. Nurses could play a vital role in boosting patient confidence and providing them with tools to improve adherence, while physicians should focus on effective communication, monitoring compliance, and adjusting treatment plans to meet individual needs. To achieve sustainable improvements in chronic disease outcomes, such as glaucoma, targeted educational programs are essential. Initiatives led by nurses can improve health literacy and patient activation, leading to better self-management and health outcomes. In resource-limited settings, interventions should include community-based education, self-management tools, peer support networks, and task-shifting models involving non-specialist providers. It is important to leverage technology, such as mobile health applications, and advocate for increased funding for chronic disease management in order to succeed, regardless of settings. Regular evaluation and adaptation of these initiatives are necessary to address the complexities of health management and meet the specific needs of diverse populations [39].

### Limitations

This study, to the best of our knowledge, is the first that examines the health literacy and patient activation levels and their potential association in patients with glaucoma in Greece. While it contributes to the existing literature, there are a few limitations worth mentioning. First, the cross-sectional design of the study did not allow causal conclusions to be drawn. Second, the study focused on a specific geographical and socioeconomic group, which may limit the generalizability of the findings to other settings and conditions. Third, the lack of a control group (patients without glaucoma) is another potential limitation regarding the generalizability of our findings. Fourth, we only included patients with specific glaucoma severity (MD > 8.5), without assessing individual disease severity in more detail. Thus, we cannot examine the potential relationships between disease progression and the studied variables. Finally, the assessment of activation and health literacy was based on self-reported data, which may contain biases. The participants may not always respond with complete accuracy or honesty. Future studies could broaden the sample range to include different geographical and socioeconomic contexts to increase the generalizability of the results. Conducting longitudinal studies would allow for the examination of causal relationships between health literacy, patient activation, and other factors. The use of objective assessment methods, combined with self-reported data, could reduce the risk of bias and provide more accurate measurements.

## 5. Conclusions

In conclusion, the findings of the present study suggest that patients with glaucoma had moderate to low levels of health literacy and patient activation. However, and surprisingly, there was no association between health literacy status and patient activation levels. Additionally, these two parameters did not show significant associations with other variables, except for education level, which exhibited a weak correlation with patients’ activation levels. Healthcare managers and policymakers could utilize our findings to improve the health literacy and patient activation of patients with glaucoma. Moreover, healthcare professionals could improve chronic disease outcomes, such as glaucoma, by providing patients with higher quality information, boosting their confidence, and encouraging them to adhere to their treatment plan. This could improve their satisfaction with care and reduce hospitalizations. Strengthening these efforts requires developing comprehensive educational programs tailored to each patient’s specific needs and circumstances, considering their health issues and cultural differences. Longitudinal studies in diverse geographical settings may provide a clearer understanding of how education level influences patient activation over time. Understanding this is crucial for future research, as it could elucidate more complex relationships and lead to improved global strategies for chronic disease management.

## Figures and Tables

**Figure 1 clinpract-15-00024-f001:**
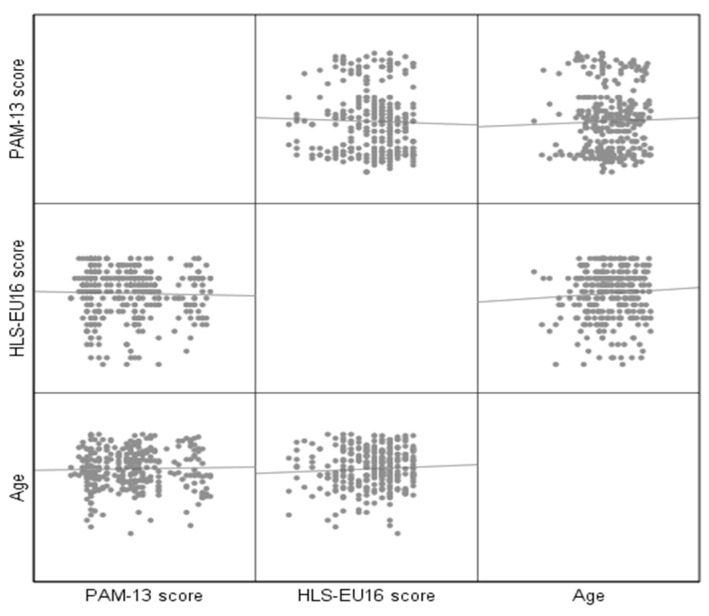
Scatter plot matrix of age, PAM-13 score, and HLS-EU16 score.

**Table 1 clinpract-15-00024-t001:** Characteristics of sample.

		Ν	N%
**Age (Years)**		**63.9 ± 14.4 ***	
Biological sex	Female	134	42.9%
	Male	178	57.1%
Education level	Primary	228	73.1%
	Secondary	44	14.1%
	Tertiary	40	12.8%
Comorbidity	No	25	8.0%
	Yes	287	92.0%
*Diabetes mellitus*		235	81.9%
*Hypertension*		230	80.1%
*Cardiovascular disease*		85	29.6%
*Cancer*		25	8.7%
*Mental disorders*		18	6.3%

(*) mean standard deviation.

**Table 2 clinpract-15-00024-t002:** Percentage distribution of participants’ responses per level of the PAM-13.

		N%
PAM-13 level	Level 1 (disengaged and overwhelmed)	67.6%
	Level 2 (becoming aware but still struggling)	9.3%
	Level 3 (taking action and gaining control)	5.4%
	Level 4 (maintaining behavior and pushing further)	17.6%

**Table 3 clinpract-15-00024-t003:** Correlations between PAM-13 and HLS-EU16.

		PAM-13 Score	HLS-EU16 Score
PAM-13 score	Pearson correlation	1	−0.030
	Sig. (2-tailed)		0.602
	N	312	312

**Table 4 clinpract-15-00024-t004:** Correlations between demographic factors and scales PAM-13 and HLS-EU16.

		Biological Sex	Age	Comorbidity	Education Level
PAM-13 score	Correlation coefficient	pbs = −0.028	r = 0.028	pbs = −0.039	ρ = 0.122 *
	Sig. (2-tailed)	0.624	0.617	0.492	0.031
	N	312	312	312	312
HLS-EU16 score	Correlation coefficient	pbs = 0.007	r = 0.060	pbs = −0.002	ρ = −0.075
	Sig. (2-tailed)	0.903	0.291	0.967	0.186
	N	312	312	312	312

*: Correlation is significant at the 0.05 level (2-tailed); r = Pearson coefficient; ρ = Spearman coefficient; pbs = Point-Biserial coefficient.

**Table 5 clinpract-15-00024-t005:** Spearman’s correlations between HLS, PAM, and educational level.

			Education Level	PAM-13 Score
Spearman’s rho	PAM-13 score	Correlation Coefficient	0.122 *	
		Sig. (2-tailed)	0.031	
		N	312	
	HLS-EU16 score	Correlation Coefficient	−0.075	−0.061
		Sig. (2-tailed)	0.186	0.279
		N	312	312

*: Correlation is significant at the 0.05 level (2-tailed).

**Table 6 clinpract-15-00024-t006:** Percentage distribution of responses per category of participants’ HLS scores.

		%
HLS-EU16 category	Inadequate HL	25.3%
	Problematic HL	34.9%
	Sufficient HL	39.7%

## Data Availability

The data that support the findings of this study are available from the corresponding author upon reasonable request due to privacy restrictions.

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
