# Peer review of "Exploring the Health Literacy and Patient Activation Among Patients with Glaucoma: A Cross-Sectional Study"

_clinpract, 2025, doi:10.3390/clinpract15020024_

Round 1
Reviewer 1 Report
Comments and Suggestions for Authors
The study investigates the health literacy and patient activation levels among glaucoma patients attending outpatient clinics in Greece. Using validated tools (HLS-EU-16 and PAM-13), it found that most participants had low to moderate health literacy and activation levels, with over 67% categorized as "disengaged and overwhelmed." Surprisingly, no significant correlation was observed between health literacy and patient activation levels, though a weak positive correlation with education level was noted. The study suggests the need for tailored educational interventions to improve patient outcomes.
Limitations: 1. Geographical Restriction: Data was limited to a specific population in Crete, reducing the generalizability. 2. Cross-Sectional Design: The study cannot establish causal relationships between the studied variables. 3. Self-Reported Data: Potential biases in responses could affect reliability. Recommendations: 1. Interventions Needed: Multifaceted educational programs are crucial to improving patient activation and health literacy. 2. Further Research: Larger, multicenter, and longitudinal studies could better elucidate the relationships between these factors and their impact on health outcomes.
Here are three questions:
1. Given that health literacy and patient activation were found to be uncorrelated, what other factors might mediate the relationship between these two variables in chronic disease management?
2. How can healthcare systems design effective interventions to simultaneously enhance health literacy and patient activation in glaucoma patients, especially in regions with limited resources?
3. Would longitudinal studies in diverse geographical settings provide a clearer understanding of how education level influences patient activation over time?
Comments on the Quality of English LanguageThe English quality of the study is generally clear and professional, suitable for academic purposes. However, there are a few areas where improvements could enhance clarity and readability:
Areas for Improvement
Sentence Structure: Some sentences are lengthy and complex, which might affect readability. For example:
Original: "Health literacy is not only a matter of individual skills. Still, it is also highly dependent on the accessibility of health systems, the communication skills of health professionals, and the level of complexity of health information."
Suggestion: "Health literacy depends not only on individual skills but also on health system accessibility, healthcare providers' communication skills, and the complexity of health information."
Minor Typos and Grammar: Occasionally, sentences could be streamlined to avoid redundancy or small grammatical errors. Example:
Original: "Therefore, the present study aimed to calculate the level of health literacy and activity level of patients’ activation in self-management, and to explore the link between them among patients with glaucoma in outpatient clinics."
Suggestion: "This study aimed to assess health literacy and self-management activation levels in glaucoma patients and explore the relationship between these factors."
Overall Recommendation
The English quality is functional but could benefit from light professional proofreading to streamline sentences, correct minor grammatical issues, and ensure consistent verb tenses. These improvements would enhance readability without altering the academic rigor of the manuscript.
Author Response
Response to the Reviewer #1 comments
Dear reviewer,
We would like to warmly thank you for your valuable feedback which is taken into consideration to improve our manuscript. Please find below our point-by-point responses.
Best regards,
Reviewer #1
The study investigates the health literacy and patient activation levels among glaucoma patients attending outpatient clinics in Greece. Using validated tools (HLS-EU-16 and PAM-13), it found that most participants had low to moderate health literacy and activation levels, with over 67% categorized as "disengaged and overwhelmed." Surprisingly, no significant correlation was observed between health literacy and patient activation levels, though a weak positive correlation with education level was noted. The study suggests the need for tailored educational interventions to improve patient outcomes.
Response: Thank you for your kind comments. We appreciate your feedback and are pleased that you found the study valuable. We believe that tailored educational interventions are crucial to improving patient outcomes and enhancing both health literacy and activation levels. We hope that our findings contribute to the development of more effective strategies for managing glaucoma and potentially other chronic conditions.
Limitations: 1. Geographical Restriction: Data was limited to a specific population in Crete, reducing the generalizability. 2. Cross-Sectional Design: The study cannot establish causal relationships between the studied variables. 3. Self-Reported Data: Potential biases in responses could affect reliability. Recommendations: 1. Interventions Needed: Multifaceted educational programs are crucial to improving patient activation and health literacy. 2. Further Research: Larger, multicenter, and longitudinal studies could better elucidate the relationships between these factors and their impact on health outcomes.
Response: Thank you for your kind comments. We have carefully considered your suggestions. We acknowledge the limitations of our study, including the geographical restriction, cross-sectional design, and reliance on self-reported data. We agree that addressing these limitations through multifaceted educational interventions is essential for improving patient activation and health literacy. Additionally, we recognize the need for further research through larger, multicenter, and longitudinal studies to gain a deeper understanding of the relationships between these factors and their impact on health outcomes. We have made the necessary adjustments to the manuscript in the implications paragraph and limitations section of the discussion, to reflect these points.
Here are three questions:
- Given that health literacy and patient activation were found to be uncorrelated, what other factors might mediate the relationship between these two variables in chronic disease management?
Response: Thank you for your kind comments. We agree that while health literacy and patient activation were found to be uncorrelated, other factors could mediate their relationship in chronic disease management. Consequently, we have incorporated the following additional information into our revised introduction section (p.2, lines 80-92).
“Although health literacy and patient activation are uncorrelated, various factors may mediate the relationship between these variables in chronic disease management. These factors may influence either the understanding of health-related information or the willingness of patients to actively participate in their care. Specifically, socio-economic factors such as educational attainment and income significantly impact the understanding and utilization of health information [9,10]. Moreover, self-efficacy and psychological conditions, including anxiety or depression, are closely associated with patients' readiness to assume an active role in their healthcare [11]. Additionally, communication with healthcare professionals and support from social networks can enhance patient engagement by providing a supportive environment for decision-making [12]. Lastly, access to health technologies and digital applications can empower patients, whereas information overload may act as a barrier [13]. Understanding these mediating factors is critical for developing tailored interventions in chronic disease management.”
- How can healthcare systems design effective interventions to simultaneously enhance health literacy and patient activation in glaucoma patients, especially in regions with limited resources?
Response: Thank you for your kind comment. We concur that regions with limited resources may encounter significant difficulties in enhancing these variables. Consequently, we have incorporated the following additional information into our revised implication section (p. 10, lines 339-356): “These findings have significant implications for both patients and healthcare professionals. It is important for healthcare professionals to actively encourage patients to engage in their own care, seek information about their condition, and participate in decision-making. Overcoming challenges related to treatment adherence is crucial, and this requires personalized educational interventions that take into account the patient’s needs, cultural background, and level of understanding [39,40]. Nurses could play a vital role in boosting patient confidence and providing them with tools to improve adherence, while physicians should focus on effective communication, monitoring compliance, and adjusting treatment plans to meet individual needs. To achieve sustainable improvements in chronic disease outcomes, such as glaucoma, targeted educational programs are essential. Initiatives led by nurses can improve health literacy and patient activation, leading to better self-management and health outcomes. In resource-limited settings, interventions should include community-based education, self-management tools, peer support networks, and task-shifting models involving non-specialist providers. It is important to leverage technology, such as mobile health applications, and advocate for increased funding for chronic disease management to succeed, regardless of setting. Regular evaluation and adaptation of these initiatives are necessary to address the complexities of health management and meet the specific needs of diverse populations [39].”
- Would longitudinal studies in diverse geographical settings provide a clearer understanding of how education level influences patient activation over time?
Response: Thank you for your kind comment. We agree that longitudinal studies could provide a clearer understanding. Therefore, we have added the following in the conclusions section (p. 10, lines 372-375): “Conducting longitudinal studies would allow for the examination of causal relationships between health literacy, patient activation, and other factors. The use of objective assessment methods, combined with self-reported data, could reduce the risk of bias and provide more accurate measurements”
The English quality of the study is generally clear and professional, suitable for academic purposes. However, there are a few areas where improvements could enhance clarity and readability:
Response: Thank you for your kind comments. We appreciate your feedback and have carefully reviewed the areas you mentioned. In response, we have made revisions to improve clarity and readability. Specifically, we have refined sentence structures, removed any ambiguous terms, and enhanced the flow of information to ensure the content is more easily understood by the reader. We believe these revisions will improve the overall quality of the study and make it more accessible for academic purposes.
Sentence Structure: Some sentences are lengthy and complex, which might affect readability.
Response: Thank you for your kind comments. We have revised the text to simplify sentence structures and improve readability, ensuring that the content remains clear and concise.
For example:
Original: "Health literacy is not only a matter of individual skills. Still, it is also highly dependent on the accessibility of health systems, the communication skills of health professionals, and the level of complexity of health information."
Suggestion: "Health literacy depends not only on individual skills but also on health system accessibility, healthcare providers' communication skills, and the complexity of health information."
Response: Thank you for your kind comments. We have revised the sentence as suggested and modified it (p2, lines 59-61 ) to improve clarity and conciseness.
Minor Typos and Grammar: Occasionally, sentences could be streamlined to avoid redundancy or small grammatical errors.
Response: Thank you for your kind comments. We have carefully reviewed the manuscript and made the necessary corrections to address the minor typos and grammatical errors. We have streamlined sentences to enhance clarity and ensure a smoother flow of ideas. Your feedback has been invaluable in improving the overall quality of the manuscript.
Example:
Original: "Therefore, the present study aimed to calculate the level of health literacy and activity level of patients’ activation in self-management, and to explore the link between them among patients with glaucoma in outpatient clinics."
Suggestion: "This study aimed to assess health literacy and self-management activation levels in glaucoma patients and explore the relationship between these factors."
Response: Thank you for your kind comments. We have revised the sentence as suggested and modified (p1, lines 21-23) to improve clarity and conciseness.
Overall Recommendation
The English quality is functional but could benefit from light professional proofreading to streamline sentences, correct minor grammatical issues, and ensure consistent verb tenses. These improvements would enhance readability without altering the academic rigor of the manuscript.
Response: Thank you for your kind comments. We appreciate your feedback and have carefully reviewed the manuscript to address the issues you mentioned. We have made revisions to streamline sentences, correct minor grammatical errors, and ensure consistent verb tenses throughout the document, thereby enhancing its readability while maintaining academic rigor.
Reviewer 2 Report
Comments and Suggestions for Authors
1. The abstract contains 297 words: you should reduce it to 250.
2. And it should contain this sentence from the introduction: In glaucoma early and effective self-management of the disease outside hospitals can slow the progression of the disease and reduce its negative impact on patients' daily lives. This includes adherence to medication, early recognition of symptoms, regular visits to the ophthalmologist, and proper use of eye drops.
3. Add in the abstract after Health literacy :requires patients to be able to find, understand, and use relevant health information
4. These two sentences can be removed from the conclusions: Multifaceted interventions could be essential for enhancing patients’ activation. Further research involving larger sample sizes, multi-center studies, and interventional design is needed.
5. In material and methods, it should be defined what comorbidity includes.
6. Results: This variable could perhaps be subdivided (into some types, Diabetes mellitus, arterial hypertension...) because in its sample 92% have comorbidity (n= 287), compared to 8% without comorbidity (n= 25). The second shows a low n less than 30.
7. In the results: the severity of glaucoma should have been included and this was as easy as including the value of the mean defect as a risk variable. This would allow us to understand whether, as the disease worsens, the patients' knowledge increases and there is an increase in enhancing patients' activation.
8. Discussion: Among the limitations of the study, the lack of a control group (patients without glaucoma) should be included.
9. This paragraph should change its conclusions and be included in the discussion as areas for improvement: Furthermore, healthcare professionals could improve chronic disease outcomes, such as glaucoma by providing patients with higher quality of information, boosting their confidence, and encouraging them to adhere to their treatment plans, this could improve their satisfaction with care and reduce hospitalizations. Furthermore, nurse-led patient education intervention could be implemented to enhance the patient activation and health literacy of patients with glaucoma in outpatient clinics. Such interventions could promote self-management and self-care behaviors, which are crucial for improved healthcare outcomes.
Author Response
Dear reviewer,
We would like to warmly thank you for your valuable feedback which is taken into consideration to improve our manuscript. Please find below our point-by-point responses.
Best regards,
Reviewer #2:
- The abstract contains 297 words: you should reduce it to 250.
Response: Thank you for your kind comments. We have reduced the length of the introduction to 247 words.
- And it should contain this sentence from the introduction: In glaucoma early and effective self-management of the disease outside hospitals can slow the progression of the disease and reduce its negative impact on patients' daily lives. This includes adherence to medication, early recognition of symptoms, regular visits to the ophthalmologist, and proper use of eye drops.
Response: Thank you for your kind comments. We appreciate your valuable feedback and have incorporated the suggested sentence into the introduction section (p.1, lines 15-20): “Glaucoma is one of the leading causes of blindness that can be mitigated through early recognition and effective management. Specifically, early and effective self-management outside hospitals can slow disease progression and reduce its negative daily impact. This includes adherence to medication, high levels of health literacy (requires patients to be able to find, understand, and use relevant health information), early recognition of symptoms, regular visits to ophthalmologists, etc.”
- Add in the abstract after Health literacy : requires patients to be able to find, understand, and use relevant health information
Response: Thank you for your kind comments. We have incorporated your suggestion into the abstract section (p.1, lines 18-19): "Health literacy requires patients to be able to find, understand, and use relevant health information."
- These two sentences can be removed from the conclusions: Multifaceted interventions could be essential for enhancing patients’ activation. Further research involving larger sample sizes, multi-center studies, and interventional design is needed.
Response: Thank you for your kind comments. We agree with your suggestion and have removed the mentioned sentences from the conclusions.
- In material and methods, it should be defined what comorbidity includes.
Response: Thank you for your kind comments. We have clarified the definition of comorbidity in the Materials and Methods section(p.3, lines 107-112 “In the present study, the dependent variables were the level of health literacy and the level of activity of glaucoma patients, while the independent variables included biological sex, age, educational level, and comorbidities (i.e., the presence of any additional chronic health conditions that may affect the management and progression of glaucoma), including diabetes mellitus, hypertension, cardiovascular diseases, cancer, and psychological disorders”.
- Results: This variable could perhaps be subdivided (into some types, Diabetes mellitus, arterial hypertension...) because in its sample 92% have comorbidity (n= 287), compared to 8% without comorbidity (n= 25). The second shows a low n less than 30.
Response: Thank you for your kind comment. We appreciate your suggestion regarding the subdivision of the comorbidity variable. Indeed, the high prevalence of comorbidity (92%, n=287) compared to the small sample of individuals without comorbidity (8%, n=25) underscores the importance of detailed analysis. We would like to inform you that we have already included the subdivision of comorbidities in Table 1 on page 5, categorizing them into distinct types such as Diabetes Mellitus, arterial hypertension, and others. We hope this approach provides a more comprehensive understanding of the data. Thank you for your valuable input.
- In the results: the severity of glaucoma should have been included and this was as easy as including the value of the mean defect as a risk variable. This would allow us to understand whether, as the disease worsens, the patients' knowledge increases and there is an increase in enhancing patients' activation.
Response: Thank you for your kind comments. In our study, the inclusion criterion used was MD >8.5 (p 3, lines 115-117), which limited our ability to include more detailed data on the severity of glaucoma on a per-patient basis. We acknowledge that the lack of a detailed record of data for each patient constitutes a limitation, which could affect the depth of the analysis and the connection between disease progression and the variables studied. (p10, lines 365-367. This limitation will be explicitly mentioned in the relevant section of the study to clarify the design constraints and enhance the transparency of the research.
- Discussion: Among the limitations of the study, the lack of a control group (patients without glaucoma) should be included.
Response: Thank you for your kind comments. We have acknowledged this limitation in the discussion section of the manuscript ((p.10, lines 364-365): “Third, the lack of a control group (patients without glaucoma) is another potential limitation regarding the generalizability of our findings.”
- This paragraph should change its conclusions and be included in the discussion as areas for improvement: Furthermore, healthcare professionals could improve chronic disease outcomes, such as glaucoma by providing patients with higher quality of information, boosting their confidence, and encouraging them to adhere to their treatment plans, this could improve their satisfaction with care and reduce hospitalizations. Furthermore, nurse-led patient education intervention could be implemented to enhance the patient activation and health literacy of patients with glaucoma in outpatient clinics. Such interventions could promote self-management and self-care behaviors, which are crucial for improved healthcare outcomes.
Response: Thank you for your kind comments. We have revised the implications paragraph of the discussion and moved/included the aforementioned part as suggested and moved it to the discussion section (p.10, lines 339-356): “These findings have significant implications for both patients and healthcare professionals. It is important for healthcare professionals to actively encourage patients to engage in their care, seek information about their condition, and participate in decision-making. Overcoming challenges related to treatment adherence is crucial, and this requires personalized educational interventions that take into account the patient’s needs, cultural background, and level of understanding [39,40]. Nurses could play a vital role in boosting patient confidence and providing them with tools to improve adherence, while physicians should focus on effective communication, monitoring compliance, and adjusting treatment plans to meet individual needs. To achieve sustainable improvements in chronic disease outcomes, such as glaucoma, targeted educational programs are essential. Initiatives led by nurses can improve health literacy and patient activation, leading to better self-management and health outcomes. In resource-limited settings, interventions should include community-based education, self-management tools, peer support networks, and task-shifting models involving non-specialist providers. It is important to leverage technology, such as mobile health applications, and advocate for increased funding for chronic disease management to succeed, regardless of setting. Regular evaluation and adaptation of these initiatives are necessary to address the complexities of health management and meet the specific needs of diverse populations [39].”
Reviewer 3 Report
Comments and Suggestions for Authors
The study entitled as “Exploring the health literacy and patient activation levels 2 among patients with glaucoma: A cross-sectional study” is well written scientific communication. The authors used two questionnaires (PAM-13 and HLS-EU-16) to determine the activation level in self-management and health literacy in glaucoma patients. They observed low to moderate level of these two factors in glaucoma patients however no association between these two factors. I only have one suggestion for the present study.
It will be interesting to know the ethnicity of enrolled glaucoma patients and their education level (if its recorded) and further analyze the data if there is any relationship between different variables.
Author Response
Dear reviewer,
We would like to warmly thank you for your valuable feedback which is taken into consideration to improve our manuscript. Please find below our point-by-point responses.
Best regards,
Reviewer #3:
The study entitled as “Exploring the health literacy and patient activation levels 2 among patients with glaucoma: A cross-sectional study” is well written scientific communication. The authors used two questionnaires (PAM-13 and HLS-EU-16) to determine the activation level in self-management and health literacy in glaucoma patients. They observed low to moderate level of these two factors in glaucoma patients however no association between these two factors. I only have one suggestion for the present study.
Response: Thank you for your kind comments and for recognizing the effort we put into our study. We deeply appreciate your feedback to further enhance the quality of this study.
It will be interesting to know the ethnicity of enrolled glaucoma patients and their education level (if its recorded) and further analyze the data if there is any relationship between different variables.
Response: Thank you for your thoughtful comments. We greatly appreciate your interest in exploring the relationships between ethnicity, education level, and other variables in our study. All participants in our study were of Greek ethnicity, following the study’s inclusion and exclusion criteria. Consequently, ethnicity was not analyzed as a variable. Regarding education level, it was recorded and analyzed concerning both the PAM-13 and HLS-EU16 scales. As detailed in Table 5 (p. 5), a statistically significant, albeit low, positive correlation was observed between education level and the PAM-13 score (Spearman’s ρ = 0.122, p = 0.031). This finding indicates that higher education levels are associated with greater patient activation. However, no significant correlations were found between HLS and educational level (p>0.05). These findings suggest that, while education level may influence patient activation, its association with health literacy was not significant in our sample. The correlations between educational level, PAM-13 scores, and HLS-EU16 scores have been incorporated into the Discussion section to further analyze the study's findings and enhance the understanding of the relationships among educational attainment, patient activation, and health literacy. (p. 9, lines 326-336). Future studies involving more diverse populations may offer deeper insights into the interactions among these variables and provide a broader perspective on the impact of ethnicity and education on health outcomes.
Round 2
Reviewer 2 Report
Comments and Suggestions for Authors
The corrections requested by the reviewer have been made.